# Are Putative Beta-Lactamases Posing a Potential Future Threat?

**DOI:** 10.3390/antibiotics14111174

**Published:** 2025-11-20

**Authors:** Patrik Mlynarcik, Veronika Zdarska, Milan Kolar

**Affiliations:** Department of Microbiology, Faculty of Medicine and Dentistry, Palacky University Olomouc, Hnevotinska 3, 77515 Olomouc, Czech Republic; veronika.zdarska@upol.cz

**Keywords:** antimicrobial resistance, beta-lactamases, horizontal gene transfer, public health

## Abstract

Background: Antimicrobial resistance is a growing global health threat, with beta-lactamases playing a central role in resistance to beta-lactam antibiotics. Building on our previous survey of 2340 putative beta-lactamases, we conducted an in-depth analysis of 129 prioritized candidates (70–98.5% amino acid identity to characterized enzymes) detected in 102 bacterial genera across 13 phylogenetic classes from environmental, animal, and human sources worldwide. Methods: We applied a motif-centric assessment of class-defining catalytic residues, evaluated the genomic context using a heuristic Index of Proximal Mobility (IPM) derived from the two immediately adjacent open reading frames, and examined the phylogenetic placement. AI-based substrate predictions were generated at a restricted scope as exploratory evidence. Results: Candidates spanned all Ambler classes (A–D); preservation of catalytic motifs was common and consistent with potential catalytic activity. Twelve of 129 (9.3%) loci had nearby mobile-element types (e.g., insertion sequences, integrases, transposases) and scored High IPM, indicating genomic contexts compatible with horizontal gene transfer. We also observed near-identical class A enzymes across multiple genera and continents, frequently adjacent to mobilization proteins. Conclusions: We propose a reproducible, bias-aware, early warning framework that prioritizes candidates based on motif integrity and mobility context. The framework complements existing surveillance (GLASS/EARS-Net) and aligns with a One Health approach integrating human, animal, and environmental reservoirs. Identity thresholds and IPM are used for inclusion and contextual prioritization, rather than as proof of function or mobility; AI-based predictions serve as hypothesis-generating tools. Experimental studies will be essential to confirm enzymatic activity, mobility, and clinical relevance.

## 1. Introduction

Antimicrobial resistance, a looming and urgent global public health threat, is expected to escalate in severity. Once confined to healthcare settings, antibiotic-resistant bacteria have now proliferated in diverse environments, including washing machines [1], urban beaches [2], wastewater treatment plants [3], post-cleaning hospital environments [4], heavy metal environments [5], city bird droppings [6], incinerators and landfills [7], dust [8], wild dolphins [9], household pets [10,11], bedbugs [12], insects and spiders [13], and microplastics [14]. This widespread emergence of antibiotic-resistant microbes poses a significant risk to a larger population. A recent study in permafrost [15] even found bacteria with a high proportion of genes encoding beta-lactamases in their genomes, further emphasizing the immediate need for global monitoring and research.

Currently, 361 beta-lactamase types have been described worldwide according to the Beta-Lactamase DataBase (BLDB), a comprehensive and widely recognized repository of beta-lactamase information (http://bldb.eu; last accessed on 30 May 2025). In this context, more than 7280 beta-lactamase genes have been described in cultured bacteria, and new variants are emerging worldwide [16]. However, new beta-lactamases are constantly being discovered.

In our previous study [17], in silico analysis of over 850 bacterial genomes from clinical and environmental isolates revealed more than 2340 candidate beta-lactamases in over 673 bacterial genera. In the present study, we focused more on candidate beta-lactamases with 70% or greater amino acid sequence identity with known beta-lactamases. Here, 70% is an inclusion bound to retain family-level homologs; a putative designation requires the preservation of catalytic motifs. We do not claim to have discovered biochemically validated enzymes. Instead, we provide a bias-aware, reproducible, early-warning framework that integrates motif integrity, periplasmic targeting, phylogenetic placement and genomic mobility to prioritize putative beta-lactamases for One Health surveillance and downstream validation.

In this study, we adopt a One Health perspective—emphasizing coordinated surveillance across human, animal, and environmental compartments—to propose sentinel sampling points and genetic markers complementing phenotypic surveillance systems such as the Global Antimicrobial Resistance Surveillance System (GLASS). This integrative approach aims to enhance early-warning capabilities and support global efforts to monitor and mitigate the spread of antibiotic resistance.

This study extends our earlier work [17] not by increasing totals, but by conducting a deeper analysis of 129 prioritized candidates drawn from the previously reported set of 2340 putative beta-lactamases. The present article operationalizes prioritization via (i) class-aware, motif-centric criteria for putative status (beyond percentage identity alone), (ii) an auditable genomic-context rubric (Index of Proximal Mobility, IPM) that summarizes nearby mobile genetic elements (MGEs) without assuming causality, and (iii) a bias-aware framing designed to complement One Health surveillance (GLASS/EARS-Net) with transparent shortlists for targeted verification.

Throughout this study, “type” refers to BLDB families (e.g., TEM, SHV, CTX-M, OXA sublineages, KPC, NDM, VIM, and IMP) rather than allelic variants. The term “361 beta-lactamase types”—the number 361 was obtained by counting all families across the Ambler classes described in the BLDB (http://www.bldb.eu:4567/, last accessed on 30 May 2025) for which at least one variant with a known GenBank accession number is available.

## 2. Results and Discussion

One hundred twenty-nine beta-lactamases in this study exhibited ≥70% amino acid identity with known beta-lactamases. The distribution of candidate beta-lactamases across various regions and hosts is shown in Figure 1. In total, we identified 102 bacterial genera and one Candidatus *Ornithobacterium* species (see Appendix A), comprising the following phylogenetic classes (*Actinomycetes*, *Alphaproteobacteria*, *Bacilli*, *Bacteroidia*, *Betaproteobacteria*, *Chitinophagia*, *Chlorobiia*, *Cyanophyceae*, *Deltaproteobacteria*, *Epsilonproteobacteria*, *Flavobacteriia*, *Gammaproteobacteria*, and *Sphingobacteriia*), in which the following candidate beta-lactamases of class A (46 times), class B (26 times), class C (12 times), and class D (45 times) were detected.

Interestingly, candidate beta-lactamase genes with ≥70% identity were typically found alone within their local genomic contexts or gene sequences, even though other, less similar or already known beta-lactamases could also be present elsewhere in the genome, contig, or sequence fragment. However, in two instances (Appendix A), two candidate beta-lactamases were located directly adjacent, suggesting a potential tandem arrangement. This co-localization’s functional or evolutionary significance remains unclear but may indicate a unique genetic organization or co-regulation.

The highest numbers of candidate beta-lactamases were found in *Chryseobacterium* spp., *Enterobacter* spp., *Bacillus* spp. and *Chitinophaga* spp., in which six, four, and three enzymes (twice), respectively, were detected. For example, two candidate class A beta-lactamases (83.2% and 80.5% amino acid identity with CGA-1) and three class B beta-lactamases (87.9% identity with ESP-1; 71.6% and 70.7% identity with GOB-like enzymes) were detected in *Chryseobacterium* spp. Furthermore, a class D beta-lactamase identified in *Chryseobacterium bernardetii* isolated in China (GenBank accession number: NZ_JACLCV010000028; GenPept ID: WP_185213878.1) showed 89% amino acid identity over 255 residues with OXA-209. The gene was located upstream of an APH(3′) family aminoglycoside O-phosphotransferase, with no additional genes identified downstream due to the contig ending at the 3′ end of the beta-lactamase, indicating a truncated genomic context. A similar OXA-209-like enzyme was also detected in *Elizabethkingia anophelis* (Taiwan; IS110 family transposase + GCN5-related *N*-acetyltransferase (GNAT); Appendix A), suggesting dissemination of this resistance determinant across multiple genera and highlighting its diversification within *Chryseobacterium* spp. Further, in *Enterobacter* spp., there were two potential class A beta-lactamases (89.1% and 82.5% amino acid identity with LAP-1 and PLA-6, respectively) and two class C beta-lactamases (87.9% and 94.5% identity with ACT-153 and CNE-2, respectively). In the case of *Bacillus* spp., there were two candidate beta-lactamases of class A (76.9% and 70.6% amino acid identity with BcIII-1 and BlaP-2, respectively) and class D (72.3% identity with BAT-1).

Some enzymes exhibited substantial amino acid divergence from known reference beta-lactamases (e.g., up to 29.1% from LCR-1), which may reflect the emergence of novel variants with altered substrate profiles or resistance mechanisms. These findings underscore the role of environmental and underexplored bacterial taxa as reservoirs of potentially clinically relevant resistance determinants.

We analyzed 129 potential or novel beta-lactamases, each sharing 70–98.5% amino acid identity with known enzymes (Appendix A). Seven isolates contained two potential beta-lactamase genes (five from China, two from the USA), indicating possible co-occurrence within single genomes. Additionally, three isolates from the USA each harbored a potential beta-lactamase gene. These genes shared 99.7% amino acid identity, but only 83.7% identity with the closest characterized beta-lactamase family, CfxA. Although these isolates were included individually in the functional analysis, they were counted only once in geographic and host-type summaries due to identical country of origin and isolation source (see Section 3).

The study further analyzed the presence of candidate beta-lactamases in bacterial isolates from various countries and demonstrated a notable distribution of these enzymes across different regions. China had the highest number of unique bacterial isolates carrying candidate beta-lactamase genes, with 31 detected cases. The USA reported 14 isolates, Japan had six, and Germany and South Korea reported five isolates each. Twenty-six isolates originated from unspecified countries. Additionally, we explored the presence of candidate beta-lactamases across different isolation environments. Most isolates were recovered from environmental samples (70), followed by 19 with an undetermined origin. Additionally, 14 were associated with animal hosts and 10 with human hosts, consistent with an environmental reservoir underpinning clinical emergence.

To better understand their functional potential, we examined conserved motifs characteristic of each class. We identified six conserved residues across the 46 candidate class A beta-lactamases, corresponding to the canonical SxxK motif, a partially conserved SDx motif (with variability at the third position), and a conserved glycine in the region typically associated with the KTG motif, which in our dataset appeared as a more variable xxG motif. Additionally, a conserved glutamate was observed in the omega (Ω) loop. While the SDx and xxG motifs exhibited variability in their flanking residues, the key catalytic residues—serine, aspartate, and glycine—were consistently preserved, indicating that the core catalytic architecture remains intact, mainly despite sequence-level divergence.

In the 26 candidate class B beta-lactamase subfamily members, key zinc-binding residues such as His116, Asp118, His179, and His240 were highly conserved. While His114 was present in the majority of sequences, substitutions with glutamate (E), asparagine (N), and glutamine (Q) were observed in six sequences, indicating some variability at this position. Cys198 was present in 16 sequences but absent in 10, suggesting possible subclass-specific differences.

The 12 uncharacterized class C beta-lactamase homologs displayed conserved SxxK, YA(S)N, and KTG motifs and characteristic sequences within the Ω- and R2-loops. Finally, several conserved motifs were identified among the 45 class D beta-lactamase candidates, though many showed notable sequence variability. For instance, the SxV motif appeared as a more variable Sxx motif with only the serine conserved, and the YGN motif was not strictly conserved, showing substitutions at all positions. Other motifs, such as STxK and KxG, were more conserved, while motifs like FxxW, FWL, and GWxxGW exhibited partial or minimal conservation, often retaining only select residues. For a detailed overview of the conserved motifs and observed sequence variations across the analyzed beta-lactamase candidates, see Appendix A.

In addition to sequence identity, we assessed the presence of conserved catalytic motifs and structural domains characteristic of each beta-lactamase class. For the orthogonal homology context, we also report the HMMER best-hit E-values (threshold: 1 × 10^−10^) for each candidate in Appendix A; lower E-values indicate more substantial similarity. These values support family-level relatedness but are not interpreted as evidence of catalytic function. This framework enables us to evaluate functional potential beyond identity thresholds: ≥70% identity serves only as an inclusion criterion to retain family-level homologs, whereas functional plausibility is assessed by the preservation of class-defining catalytic residues, independent of global identity. Accordingly, conservation of these motifs supports a hypothesis-generating designation of putative beta-lactamases rather than a functional claim, and is consistent with potential catalytic activity. Nevertheless, future experimental validation will be essential to confirm enzymatic activity and clinical relevance.

The phylogenetic analysis revealed that several candidate beta-lactamases clustered near known representatives of established families, particularly within classes A and D. Some sequences formed distinct branches, suggesting the presence of potentially novel or highly divergent beta-lactamase families. For detailed class-specific trees and sequence information, see Appendix A.

Identification of MGEs revealed the presence of genes for transposases, integrases/recombinases, and mobilization proteins in 12 cases (Table 1), with a higher abundance of transposases and/or recombinases/integrases, which may lead to overproduction of beta-lactamases and facilitate gene transfer, spreading beta-lactamase genes. A comprehensive list of all identified candidate beta-lactamases is provided in Appendix A.

The figure illustrates the number of unique bacterial isolates harboring candidate beta-lactamase genes, categorized by geographic origin and source type. Each “×” represents one unique isolate carrying at least one candidate beta-lactamase gene; isolates with multiple genes are counted only once (see Section 3). The inset table summarizes source categories and IPM classes by region. Created with BioRender.com. A focused analysis of 12 putative beta-lactamase sequences identified in this study and associated with MGEs revealed various structural and functional features across class A–D representatives. All analyzed sequences retained key conserved catalytic residues and class-specific motifs, albeit with some variations, supporting their potential enzymatic activity. The class A enzymes preserved the SxxK, SDN, and Ω-loop motifs, with partial conservation of the KTG motif. The class B enzyme retained all essential residues for metallo-beta-lactamase activity. Class C representatives showed conservation of SxSK, YSN, and KTG motifs. Class D variants exhibited conserved STFK, KTG, and GWxxGx motifs, with variable conservation in FxxW, SxV, YGN, and FWL motifs, suggesting functional potential despite sequence divergence. Notably, upstream or downstream MGEs—including IS3, IS30, IS91, IS110, IS3/IS911 transposases, and tyrosine-type recombinases—suggest a high potential for horizontal gene transfer. These beta-lactamase genes were located near mobilization proteins or other resistance-associated elements in several cases, indicating a genomic context conducive to mobility and dissemination.

In addition, two nearly identical class D beta-lactamases (89% amino acid identity over 255 residues with OXA-209) were identified in different bacterial genera and geographic locations. The first enzyme (UKY84509.1) was found in *Elizabethkingia anophelis* isolated from a human host in Taiwan (GenBank accession: CP077754), flanked by an IS110 family transposase and a GNAT family N-acetyltransferase, suggesting potential mobility. Interestingly, the same enzyme (WP_185213878.1) was also detected in *Chryseobacterium bernardetii* from a human host in China (GenBank accession: NZ_JACLCV010000028), but without association to a transposase; instead, it was flanked by an APH(3′) family aminoglycoside O-phosphotransferase. This observation may indicate horizontal gene transfer and genomic integration into distinct genetic contexts across species and regions.

Under our IPM scheme, loci with High IPM (≥2 distinct MGE types) are prioritized for follow-up as they summarize genomic contexts compatible with mobility. This is a heuristic prioritization, not proof of mobilization or clinical impact. Further details on the structural features, conserved motifs, predicted substrate profiles, and genetic contexts of these candidate beta-lactamases associated with MGEs are provided in Appendix A.

PlasmidFinder [18] analysis identified plasmid-associated replicon sequences in five of the 129 genomes analyzed (Appendix A). However, none of these replicons were located on the same contigs as the *bla* genes or in proximity to MGEs identified in our in silico analysis. In one isolate with a complete circular genome, the distance between the identified replicon and the nearest MGE exceeded 1 Mb, further limiting evidence for plasmid-mediated mobilization. In contrast, most other assemblies consisted of fragmented whole-genome sequencing data, which inherently limited the resolution of genomic context and the ability to assess physical linkage between *bla* genes, MGEs, and plasmid replicons. Additionally, PlasmidFinder applies primarily to Gram-positive bacteria and members of Enterobacterales, excluding many Gram-negative non-Enterobacterales isolates in our dataset. For loci where no replicon was detected or where replicon sequences were distant from *bla* genes, an IPM classification of Low was assigned. However, in cases where MGEs were detected but PlasmidFinder yielded no results, the true mobilization potential remains uncertain due to these methodological limitations.

Our detailed study of candidate beta-lactamases binding to MGEs resulted in a significant discovery. Specifically, we identified two candidate class A beta-lactamases with identical amino acid sequences in *Enterobacter* spp. (EJF29815.1) and *Cedecea* spp. (WP_008460137.1), both of which were found in mosquitoes and had an insertion sequence (IS) located nearby. This finding is crucial as it highlights potential sources of antibiotic resistance, with one case originating from the USA and the other from an unspecified location.

An IS, a short DNA segment that acts as a simple transposable element, was also found around candidate class C beta-lactamase described in *Aeromonas veronii* (BBT97416.1) isolated from a wastewater treatment plant effluent in Japan. This IS has also been detected in other bacterial species of this genus, such as *Aeromonas caviae* (GenBank accessions: AP022242, AP019196), all isolated from the same source and country.

The most worrying situation seems to be related to candidate class A beta-lactamases, which show 99.9% nucleotide and 99.7–100% amino acid identity (change in one amino acid) with each other and have been detected in *Bacteroides faecis* (UVS32091.1) and *Parabacteroides distasonis* (UVS67385.1); these enzymes were located near the mobilization protein. More detailed in silico analysis indicated their occurrence in other species of the above genera, such as *Alistipes* spp. (GenBank accession: RKBL01000025), *Bacteroides ovatus* (GenBank accession: CP103071), *Bacteroides fragilis* (GenBank accessions: CP036539, CP119600), *Bacteroides salyersiae* (GenBank accession: CP072243), *Bacteroides* spp. (GenBank accession: CP040630), *Parabacteroides distasonis* (GenBank accession: CP072231), and *Parabacteroides goldsteinii* (GenBank accession: CP081906). In addition, these enzymes have also been detected in other bacterial species, such as *Escherichia coli* (GenBank accession: WKRP01000318), *Phocaeicola vulgatus* (GenBank accession: CP096965) and *Prevotella* spp. (GenBank accession: AP035786). These enzymes originated from humans and were found in India, the USA, Canada, Denmark, and the Netherlands, and from horses (*Equus caballus*) in Japan. This fact illustrates the seriousness of the situation, as the presence of beta-lactamases with a mobilization protein has been detected in many genera and species, indicating their widespread distribution and potential for the spread of antibiotic resistance.

Our study highlights the significant distribution of candidate beta-lactamases across different bacterial genera, regions, and hosts. The presence of these enzymes, with exact amino acid identities with known beta-lactamases, in genera such as *Cedecea*, *Enterobacter*, *Alistipes*, *Bacteroides*, and *Parabacteroides*, suggests that horizontal gene transfer may play a role in spreading antibiotic resistance. This is supported by the identification of MGEs surrounding these genes, which may facilitate their transfer. China had the highest number of candidate beta-lactamases, with the majority detected in environmental samples. The detection of these enzymes in various environmental and clinical settings underscores the need for comprehensive monitoring and international surveillance. To combat antibiotic resistance, global cooperation is needed. MGEs, such as transposases, highlight the potential for these resistance genes to spread. This is crucial for predicting treatment failures and understanding the spread of resistance within and across bacterial species.

To complement the motif-based and phylogenetic annotations summarized in Appendix A, we applied an AI-driven model with the specific purpose of inferring exploratory, class-level substrate tendencies (penicillins, cephalosporins, and carbapenems) directly from sequence features and conserved motifs. For orientation, we applied the same procedure to four well-characterized beta-lactamases (CTX-M-15, VIM-1, ACC-1, and OXA-1), and the corresponding AI-generated substrate profiles are reported in Appendix A (sheet 2) to illustrate the model’s behavior on enzymes with known broad substrate spectra. These examples are intended to provide qualitative context, rather than a formal calibration or performance benchmark. The AI outputs are overall interpreted as class-level substrate hypotheses, not as clinical activity claims. This approach offers an orthogonal perspective to the motif/phylogeny-based classification and may help prioritize candidate enzymes for further functional follow-up. However, we encountered several technical inconsistencies while interpreting AI-derived predictions, including inaccurate protein lengths (even after signal peptide correction), incorrect molecular weight estimates, and misaligned amino acid numbering for both catalytic residues and conserved motifs. These issues underscore the importance of cautious interpretation and manual validation when integrating AI-based annotations with curated sequence data.

Signal peptide prediction was performed on a total of 129 candidate enzyme sequences. Prediction scores ranged from 0.1 to 1.0. Using a conservative threshold of ≥0.9 to indicate high-confidence signal peptides, 82 candidates (63.6%) met this criterion (Appendix A). In four cases, no signal peptide was detected, which may reflect either incomplete sequence data or the genuine absence of a classical signal peptide. Certain beta-lactamases—such as Class D OXA enzymes from *Acinetobacter* spp.—are known to be lipid- and membrane-bound rather than secreted into the periplasmic space [19]. These enzymes may evade detection by standard signal peptide prediction tools optimized for soluble, Sec-pathway-targeted proteins.

Among High-IPM loci (Table 1), four out of nine sequences (44.4%) carried high-confidence signal peptides, a proportion lower than in the overall set. This pattern is consistent with a greater share of non-secreted or membrane-associated enzymes within High-IPM contexts. Conversely, three Moderate-IPM enzymes showed low signal peptide likelihood (0.1), while mobilization proteins were detected nearby. These observations are compatible with horizontal transfer-related genomic neighborhoods but do not establish a mechanism or directionality. Alternative localization or functions unrelated to classical secretion remain plausible.

Compared with our earlier survey of 2340 putative beta-lactamases [17], the present work deep-dives a defined subset of 129 prioritized candidates, introducing (i) motif-centric criteria for putative status (beyond percentage identity alone), (ii) an auditable IPM that summarizes MGE types in the two flanking genes as a heuristic context for prioritization (we did not perform quantitative/enrichment testing; IPM is not statistical evidence of mobility), and (iii) an explicitly bias-aware, One Health-aligned prioritization intended to complement existing surveillance.

The presence of beta-lactamases in diverse environments suggests that natural habitats may serve as reservoirs for these resistance genes, potentially allowing them to be transferred to clinical settings. Environmental strains provide a significant reservoir of new resistance genes, including carbapenemases, which can be disseminated through the food chain, accelerating global antibiotic resistance. Human activities facilitate the spread of resistance from various reservoirs such as washing machines and urban beaches, contributing to the complexity of combating antibiotic resistance. This broad distribution underscores the severity of the issue and the need for further research to understand and tackle antimicrobial resistance. Given the growth of public genome/metagenome datasets and our identification of putative sequences with conserved catalytic motifs, further beta-lactamase diversity is plausible. The contribution of such enzymes to beta-lactam resistance remains to be established through experimental characterization and epidemiological evidence.

This work is in silico. Despite de-duplication and exploratory subsampling, residual dataset bias may persist. PlasmidFinder analysis identified plasmid-associated replicons in only a small subset of loci, and its applicability was limited by taxonomic scope and fragmented assemblies. Instead, our in silico analysis of flanking MGEs provided more consistent evidence of genomic contexts conducive to gene mobility. The AI-assisted substrate predictions in this study are zero-shot and exploratory, and have only been qualitatively examined on four control enzymes (CTX-M-15, VIM-1, ACC-1, and OXA-1). We did not compute confusion matrices or other formal performance metrics, and no biochemical (wet-lab) validation was performed. Accordingly, the AI outputs should be interpreted strictly as hypothesis-generating signals rather than as evidence of actual hydrolytic activity or clinical impact. HMMER E-values are included as auxiliary homology support only and are not used to infer catalytic function in the absence of conserved class-specific motifs. Accordingly, our 70% threshold serves as an inclusion heuristic, not a functional cutoff; likewise, IPM is presented as a literature-informed, heuristic context summary for prioritization, rather than statistical evidence of mobility. Functional plausibility in this study rests on motif integrity and genomic context.

Our framework complements established phenotypic surveillance systems, such as the GLASS and EARS-Net. Flagging putative alleles with preserved catalytic motifs and higher IPM scores provides a hypothesis-generating early-warning shortlist to guide targeted PCR or metagenomic screening at sentinel sites for prioritized verification. This integrative, One Health-aligned approach connects clinical, animal, and environmental compartments, enabling more focused monitoring and earlier detection of candidates of concern. We emphasize that IPM and sequence-homology measures serve as contextual priors rather than proof of mobility or clinical impact; confirmed relevance will require experimental validation and continued phenotypic surveillance. Sustained surveillance and research efforts remain crucial to combating antibiotic resistance and safeguarding public health.

## 3. Materials and Methods

A total of 125 bacterial genomes and gene sequences encoding 129 putative beta-lactamases were retrieved from the National Center for Biotechnology Information (NCBI) and analyzed. These sequences, annotated using the NCBI Prokaryotic Genome Annotation Pipeline [20], were filtered using the “Find Annotations” function in Geneious Prime 2025.1.3 [21]. The study focused on sequences exhibiting 70.0–98.5% amino acid identity with known beta-lactamases, as determined using the BLDB BLAST tool (http://www.bldb.eu:4567/, last accessed on 30 May 2025) [22].

This inclusion threshold (≥70%) was chosen to retain family-level homologs, including divergent candidates, while filtering out remote, low-similarity hits. Functional plausibility was assessed by the preservation of class-defining catalytic motifs, rather than by percentage identity. A ~70% identity boundary has also been used in large-scale antibiotic-resistance gene discovery to organize homologs at the family level (e.g., aminoglycoside-modifying enzyme families clustered at <70% identity after hidden Markov model detection in Lund et al. [23]), which is consistent with our use of ≥70% as an inclusion bound rather than a functional cutoff. For the orthogonal homology context, we also report HMMER [24] best-hit E-values (threshold: 1 × 10^−10^) in Appendix A; these scores indicate sequence relatedness but are not used as functional evidence.

Protein sequences were subjected to in silico structural and functional analysis using Geneious Prime, which enabled multiple sequence alignments, identification of conserved catalytic residues, and annotation of class-specific motifs such as SxxK, SDN, KTG, and HxHxD. The multiple sequence alignment was performed using MUSCLE 5.1 [25]. Structural features [26], including the Ω- and R2-loops and their possible extents [27], were also examined to support classification into beta-lactamase classes A–D [28]. The presence of conserved domains and motifs was used to infer enzymatic functionality and distinguish between active enzymes and pseudogenes. Sequences retaining key catalytic residues and class-defining motifs were classified as putative beta-lactamases. At the same time, those with disrupted or missing essential features were flagged as unlikely functional, regardless of global sequence identity.

A maximum-likelihood phylogenetic tree was constructed using PhyML (Le-Gascuel model, 100 bootstrap replicates) in Geneious Prime to assess the evolutionary relationships of candidate beta-lactamases [29]. The tree included one representative from each beta-lactamase family described in the BLDB and the candidate sequences identified in this study. Class-specific trees (Appendix A) were rendered in Geneious Prime as circular, unrooted cladograms (Transform branches: cladogram; Order branches: increasing). Branch labels were set to bootstrap proportions with “Show next to node” enabled; labels were exported as whole-number percentages. For readability, putative beta-lactamase candidates (“P-”) are in bold and known beta-lactamase families are in light gray.

Claude 3.7 Sonnet (Anthropic), an advanced AI platform [30], was employed to obtain qualitative predictions of likely beta-lactam substrate classes for each enzyme. For each full-length amino acid sequence, the model was queried to integrate sequence-based features, motif composition, and similarity to known beta-lactamases, and to suggest broad substrate profiles, including potential activity against penicillins, cephalosporins, and carbapenems. We did not attempt to infer allele-specific hydrolytic rates, minimum inhibitory concentration shifts, or kinetic parameters; the outputs are therefore interpreted as class-level substrate tendencies, not as quantitative or clinically actionable activity profiles. To illustrate the behavior of the model on enzymes with known properties, we applied the same procedure to four well-characterized beta-lactamases (CTX-M-15, VIM-1, ACC-1, and OXA-1) and report the resulting AI-generated substrate profiles in Appendix A (sheet 2), where they can be qualitatively compared with their established phenotypes. This AI-assisted approach was used as an exploratory tool to support interpretation of sequence–function relationships, particularly for enzymes with substantial divergence from known reference sequences.

We predicted N-terminal secretion signals to assess periplasmic targeting of candidates (SignalP 6.0) [31]. Additionally, PlasmidFinder 2.1 was used to indicate the plasmid origin of *bla* genes [18]. The minimum coverage threshold was set to 60% and the minimum identity threshold to 80%.

To assess the potential for horizontal gene transfer of beta-lactamase genes, we analyzed their genomic context, focusing on the regions flanking each *bla* locus (*n* = 129). Annotated sequences were manually inspected using Geneious Prime to identify MGEs, including IS, integrases, recombinases, transposases, and mobilization proteins. The presence and orientation of these elements were confirmed through manual curation. Additional metadata, including the host organism, source of isolation, and geographic origin, were obtained from the NCBI Protein database and integrated into the analysis to provide a comprehensive view of the genomic and ecological context of beta-lactamase genes.

To quantify locus-level mobility potential, we applied an IPM based on the two immediately adjacent open reading frames (ORFs) flanking each *bla* locus (one upstream and one downstream, where available). Mobilization proteins were assigned a weight of 1, whereas IS, integrases, recombinases, and transposases were each assigned a weight of 2. Each element type was counted once per locus (irrespective of repetition) to avoid inflation due to tandem copies or fragmented contigs. Loci were categorized as Low (IPM = 0), Moderate (IPM = 1), or High (IPM ≥ 2). The weighting is a descriptive heuristic—a transparent summary of context rather than a validated predictor—and conclusions are interpreted directionally. For loci at contig edges, if only one adjacent ORF was available, IPM was computed from the available side.

Rationale: We assign a higher weight to IS/integrase/transposase (2) than to mobilization proteins (1) because the former directly facilitate gene capture, integration, or transposition. In contrast, mobilization proteins indicate a conditional ability to mobilize, which requires assistance from helper conjugative systems. Alternative reasonable weights can be considered under the same type-once counting rule [32,33,34]. This rubric is fully rule-based and therefore reproducible across studies that use comparable annotation pipelines. Given gene annotations for the two flanking ORFs, any group can recompute IPM by applying the same element-type categories, type-once counting rule, and fixed weights. Because IPM depends only on the presence or absence of these element types in the flanking genes, it can be readily automated and integrated into genome- or metagenome-analysis workflows.

To reduce sampling bias from public genome repositories, we applied a multi-step selection strategy focused on minimizing redundancy and maximizing biological diversity. Near-identical isolates (ANI ≥ 99.9%) were deduplicated and only one representative genome per BioProject was retained unless functional or taxonomic differences justified inclusion. Additional sequences from the same BioProject were marked as “not applicable” for subsampling, indicating exclusion due to redundancy control. An exception was made for BioProject PRJNA224116 (RefSeq Prokaryotic Genome Annotation Project), which contains over 537,000 BioSamples; 54 unique sequences were included to reflect its diversity. Three other BioProjects—PRJNA646575, PRJDB6962, and PRJNA169065—contributed two genomes under similar considerations.

Selection rules included: (i) Single isolate count: If a genome contained multiple beta-lactamase genes, each gene was analyzed separately for functional analysis; however, the isolate was counted only once in geographic and host summaries. (ii) Identity-based grouping: Beta-lactamases with >99.5% amino acid identity found in different genera originating from the same country and source were counted once. Identical enzymes from different countries were counted separately to reflect potential horizontal gene transfer. These decisions are reflected in Figure 1, where visualization rules differ: isolates are counted once regardless of the number of beta-lactamase genes or functional redundancy.

Genomes were selected based on the presence of candidate genes, with metadata (country, host, and source) retrieved post hoc. The process was exploratory rather than strictly stratified.

## 4. Conclusions

The study prioritizes potentially novel resistance determinants that may evade current diagnostic detection by focusing on enzymes with 70–98.5% amino acid identity to known beta-lactamases. Our analysis identifies putative beta-lactamases that preserve class-defining catalytic motifs across Ambler classes A–D and occur in diverse hosts and environments. Combining motif integrity, phylogenetic placement, and genomic mobility context, we identify putative beta-lactamase candidates across A–D classes and prioritize those most likely to disseminate. Notably, 12 out of 129 (9.3%) candidates reside next to MGEs, exemplifying loci with non-negligible mobility potential and contexts compatible with horizontal gene transfer. These loci represent candidates for targeted surveillance.

We present a reproducible, bias-aware, early warning framework that prioritizes candidates based on motif integrity and genomic mobility context. Integrating structural motif analysis, AI-based substrate prediction, and phylogenetic profiling reveals that many enzymes retain conserved catalytic features, consistent with potential catalytic activity. AI-based substrate predictions were conducted at the allele level and are provided as exploratory, hypothesis-generating evidence requiring experimental validation. Functional characterization of high-priority candidates is beyond the scope of this work and will require targeted efforts by groups with access to relevant isolates; accordingly, all in silico results are presented strictly as hypothesis-generating.

These findings outline how outputs can complement existing surveillance systems such as the GLASS and EARS-Net within a One Health approach—integrating human, animal, and environmental reservoirs. The study reinforces the importance of this integrated strategy for antimicrobial resistance monitoring and provides valuable insights for anticipating emerging resistance threats and guiding public health interventions.

## Figures and Tables

**Figure 1 antibiotics-14-01174-f001:**
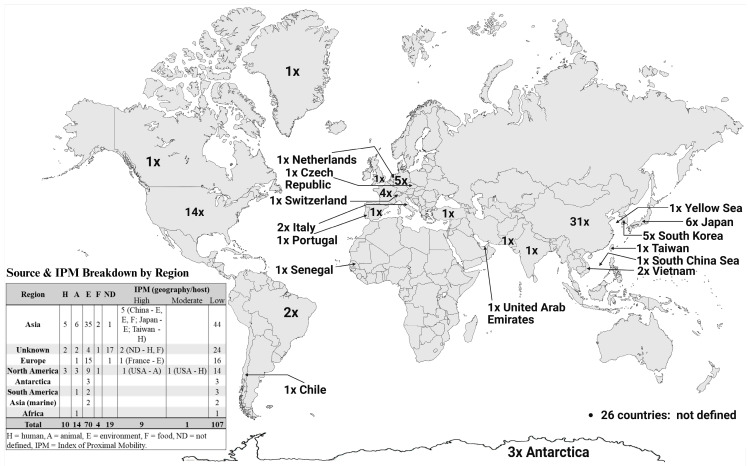
Global distribution of candidate beta-lactamase-carrying isolates across regions and host types, with IPM breakdown.

**Table 1 antibiotics-14-01174-t001:** Genomic surroundings and mobility potential of candidate beta-lactamases.

Bacterial Genera	Putative B-L Class	IPM Value/Class	Size (AA)	GenPeptID/Identity	Identified Motifs (Conserved/Variant)	Signal Peptide Likelihood
*Enterobacter*	A	2/high	285	EJF29815.1 (89.1% identity over 285 AA with LAP-1)	STFK, SDN, KTG, 93.8% AA identity (15/16) with LAP-1 across the Ω-loop	0.8
*Ornithobacterium* *	A	2/high	294	CAI9429279.1 (84.7% identity over 294 AA with RASA-1)	STFK, SDN, ETG (modified KTG), 93.8% AA identity (15/16) with RASA-1 across the Ω-loop	1
*Bacteroides*	A	1/moderate	322	UVS32091.1 (83.7% identity over 318 AA with CfxA-3)	SVFK, SDN, KTG, 81.3% AA identity (13/16) with CfxA-2 across the Ω-loop	0.1
*Parabacteroides*	A	1/moderate	322	UVS67385.1 (83.7% identity over 318 AA with CfxA-2)	SVFK, SDN, KTG, 81.3% AA identity (13/16) with CfxA-2 across the Ω-loop	0.1
*Phocaeicola*	A	1/moderate	322	USS67813.1 (83.7% identity over 318 AA with CfxA-5)	SVFK, SDN, KTG, 81.3% AA identity (13/16) with CfxA-2 across the Ω-loop	0.1
*Psychrobacter*	A	2/high	299	WP_102076509.1 (78% identity over 296 AA with CARB-5)	STFK, SDN, RTG (modified KTG), 87.5% AA identity (14/16) with CARB-5 across the Ω-loop	0.8
*Vibrio*	B	2/high	250	WP_318122257.1 (85.8% identity over 240 AA with TMB-1)	His94, His96, Asp98, His156, Cys175, and His217—all conserved in comparison with TMB-1	1
*Pseudomonas*	C	2/high	396	UVJ43626.1 (92.2% identity over 396 AA with RSC1-1)	SVSK, YSN, KTG; 98% AA identity (50/51) across Ω-loop, 80% (20/25) within R2-loop: versus RSC1-1	0.7
*Aeromonas*	C	2/high	380	BBT97416.1 (86.4% identity over 381 AA with PAC-1)	SLSK, YSN, KTG; 90.2% AA identity (46/51) across Ω-loop, 88% (22/25) within R2-loop: versus PAC-1	1
*Elizabethkingia*	D	2/high	274	UKY84509.1 (89% identity over 255 AA with OXA-209)	PASTFK, FKGW (modified FxxW), SCV (SxV), YKG (YGN), KTG, GWFVGY (modified GWxxGW); 87.5% AA identity (14/16) with OXA-209 across the Ω-loop	0.8
*Comamonas*	D	4/high	256	WP_202883143.1 (81.3% identity over 268 AA with OXA-926)	PASTFK, GDAW (modified FxxW), SVF (SxV), YGN, KTG, GWYVGW (GWxxGW); 81% AA identity (17/21) with OXA-926 across the Ω-loop	0.8
*Aromatoleum*	D	2/high	261	WRL48780.1 (70.9% identity over 258 AA with LCR-1)	STFK, ISNW (modified FxxW), SCV (SxV), YGQ (YGN), KTG, GWYVGF (modified GWxxGW); 80% AA identity (12/15) with LCR-1 across the Ω-loop	1

* Candidatus *Ornithobacterium*. B-L: beta-lactamase. AA: amino acids. E: glutamate. IPM: Index of Proximal Mobility. Ω: omega. Additional information on candidate beta-lactamases with mobile genetic elements is provided in Appendix A.

## Data Availability

The genes encoding putative beta-lactamases are available in the NCBI GenBank. The data presented in this study are available in the “Results and Discussion” section.

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
