# Peer review of "Are Putative Beta-Lactamases Posing a Potential Future Threat?"

_antibiotics, 2025, doi:10.3390/antibiotics14111174_

Round 1

Reviewer 1 Report (Previous Reviewer 2)

Comments and Suggestions for Authors

Comments and Suggestions for the Authors

The article, ‘Are putative beta-lactamases posing a potential future threat? ' Presents an interesting reading experience. Approaches and the results are very well described and easily understandable. The article can be accepted for publishing. Data collection and analysis are also acceptable

Author Response

Response to Reviewer 1

We thank the reviewer for the positive assessment of our revised manuscript and for recognizing the clarity of the approaches and results. We appreciate the recommendation for acceptance. We are grateful to the reviewer for their time and support.

Reviewer 2 Report (New Reviewer)

Comments and Suggestions for Authors

This manuscript presents an in-silico investigation of 129 presumed β-lactamases sharing 70–98.5% amino-acid identity with known enzymes, identified across 102 bacterial genera from environmental, animal, and human sources. The study integrates motif-based analysis, phylogeny, AI-assisted substrate prediction, and genomic context evaluation (including mobile genetic elements) to propose an early-warning framework for resistance surveillance. The topic is relevant for Antibiotics, fitting the scope of antimicrobial resistance, One Health, and emerging β-lactamases. The dataset is extensive, and the concept of a “bias-aware early-warning framework” is innovative. However, the manuscript currently reads more as an exploratory bioinformatic survey than a hypothesis-driven scientific study. Several issues concerning clarity, data interpretation, validation, and organization must be addressed before the work can be considered for publication.

Major Comments

  1. Abstract is overly long and should be more condensed to highlight key findings and methodology (≤250 words).
  2. The manuscript builds directly on a previous publication by the same authors (Zdarska et al., 2024). The novelty of this follow-up study should be explicitly clarified: what new biological insights or methodological advances are offered beyond extending the earlier dataset?
  3. The use of ≥70 % amino-acid identity as a cutoff for putative β-lactamases is random. The rationale for choosing this threshold needs stronger justification and comparison with established approaches (e.g., BLDB family boundaries, HMM-based classification).
  4. The authors carefully inspected flanking regions of bla genes to identify MGEs and confirmed their orientation, which is praiseworthy. However, the manuscript interprets MGE proximity as direct evidence of mobility potential without performing any quantitative or enrichment analyses. Including such an analysis, or at least discussing this limitation, would strengthen the conclusions about gene mobility.
  5. The IPM is clearly described (weights for different MGEs are provided), yet its empirical basis remains unclear. Please justify the choice of weighting values (1 vs. 2), provide any benchmarking or validation, and discuss the potential sensitivity of results to these arbitrary weights.
  6. Figures 1 and 2 are insufficiently informative: Figure 1 lacks quantitative scale or legends for host/environment distribution; Figure 2 lacks visible branch support values (displaying these values or at least indicating well-supported vs. poorly supported nodes, is essential to assess the reliability of phylogenetic clustering).
  7. Consider reorganizing Results and Discussion into clearer subsections (Distribution, Conserved Motifs, Phylogeny, Mobility, AI predictions).
  8. The tone sometimes becomes influential (“we are convinced that we will soon witness…”, line 321). Replace subjective statements with objective, evidence-based language.
  9. Although limitations are mentioned, they should be consolidated into a dedicated subsection discussing dataset bias, incomplete assemblies, lack of experimental validation, and speculative nature of AI predictions.
  10. Properly cite Figure S2 in the main text.

Author Response

We thank the reviewer for the thorough assessment and constructive guidance. Below, we address each point and indicate how the revised manuscript responds. Throughout, we maintain a conservative scope: all enzymes are treated as putative, AI outputs are hypothesis-generating, and the manuscript presents a bias-aware early-warning framework that complements existing surveillance.

Reviewer 3 Report (New Reviewer)

Comments and Suggestions for Authors
  1. Author use Claude 3.7 Sonnet (Anthropic) and it is innovative. Please, clarify if AI predictions, it is limited to class-level substrate tendencies or specific hydrolytic patterns. Consider including a confusion matrix or validation table comparing AI predictions versus known enzyme activity.
  2. Please strength the explanation of the Index of Proximal Mobility (IPM)—is it reproducible across studies? Could it be automated?
  3. The statement “AI-based substrate predictions are provided as exploratory evidence” should specify that no biochemical validation was done.

Author Response

We thank the reviewer for the supportive remarks and helpful suggestions. We have clarified the scope of the AI-based predictions, strengthened the description of the Index of Proximal Mobility (IPM), and made explicit that no biochemical validation was performed. Our revisions aim to clearly frame the AI component and IPM as reproducible, hypothesis-generating tools within a conservative, in-silico early-warning framework.

Round 2

Reviewer 2 Report (New Reviewer)

Comments and Suggestions for Authors

After the second round of review, I believe the authors have addressed the previous comments, and the manuscript is ready for publication.

Reviewer 3 Report (New Reviewer)

Comments and Suggestions for Authors

Manuscript can be accepted for publication

This manuscript is a resubmission of an earlier submission. The following is a list of the peer review reports and author responses from that submission.

Round 1

Reviewer 1 Report

Comments and Suggestions for Authors

Authors have selected betalactamase based antibiotics and predicted them as a potential threat for future apart from having drug resistant superbugs. However, the prudent use of antibiotics is necessary while the excessive use is creating a havoc and threat for future living beings.

The manuscript entitled "Are new beta-lactamases posing a potential future threat?" sounds a good contribution but there are still some concerns need to be addressed which are mentioned below:

1. How did the authors addressed the biases having a great potential during the genome selection as the data is representing the isolates from the public databases?

2. The conserved motifs mentioned in the manuscript suggests functionality which is a good idea but what are the thresholds of the sequences divergence would make a non-functional beta-lactamase?

3. Did the authors identified plasmids related to beta-lactamase genes?

4. How does the resistance gene proximity influence the mobility potential of the enzymes related to resistance genes?

5. How the findings suggested by the authors would fit existing AMR monitoring system?

6. Could the environmentally existing beta-lactamases poses direct impact on human health or they need any transmission medium to impact?

7. Any beta-lactamases inhibitors were tested in-silico against candidate enzymes to check the cross-resistance?

8. How AI could be applicable to predict emerging resistance to non-beta-lactamase based antibiotics? What models does the authors suggested it can be modified?

Reviewer 2 Report

Comments and Suggestions for Authors

Comments and Suggestions for the Authors

The article, ‘Are new beta-lactamases posing a potential future threat?' presents an average reading experience. The study is too generic; statements are completely based on predictions and predictions could be biased towards algorithms used. In my opinion authors didn’t find ‘new’ enzymes, as they claimed in the title, rather, they found ‘putative’ enzymes with potential beta-lactamases activity. Authors have emphasized that ‘urgent need for global surveillance’, which is already in place under the umbrella of WHO. Are the authors trying to suggest something else that have been missed by world organizations, and which is not already in effect? If so, authors should clarify and try establishing the facts firmly citing points from their study. In my opinion, at current state, the article is not unique/novel and cannot be considered for publishing as ‘priority’.

The following are my comments and suggestions to improve the quality of the manuscript,

  • The title should mention the term ‘putative’ as the whole study is based on predictions, no hard evidence.
  • Abstract is okay.
  • Line 29 in Abstract and line 303 in conclusion; what did the authors mean by ‘One Health’? Need clarification.
  • Key words are okay.
  • Authors should explain what they have meant by ‘361 beta-lactamase types’ and where in Beta-Lactamase DataBase (BLDB) that has been mentioned?
  • Figure 1. What does ‘X’ marks represent? Figure legends should be standing alone explanation what the figure is about. Nothing is comprehensible from the figure in current stage. Re-write figure legend.
  • Table S1. Why are some of the rows highlighted?
  • In my opinion, Figure 2 doesn’t have the merit to be in the main text, can be moved as a Supplementary image.
  • Figure 3, nothing is legible in phylogenetic trees. Each tree should be big enough to be legible. Authors could consider submitting each of them as separate image files.
  • References are okay.

Reviewer 3 Report

Comments and Suggestions for Authors

Dear Authors,

This is a well studied insilico study of candidate β-lactamase genes. However, it lacks experimental validation limits its suitability for this journal. The work may be better suited to a bioinformatics or microbial genomics–focused journal.